# Hunted Wild Boars in Sardinia: Prevalence, Antimicrobial Resistance and Genomic Analysis of *Salmonella* and *Yersinia enterocolitica*

**DOI:** 10.3390/foods13010065

**Published:** 2023-12-23

**Authors:** Giuliana Siddi, Francesca Piras, Maria Pina Meloni, Pernille Gymoese, Mia Torpdahl, Maria Fredriksson-Ahomaa, Mattia Migoni, Daniela Cabras, Mario Cuccu, Enrico Pietro Luigi De Santis, Christian Scarano

**Affiliations:** 1Department of Veterinary Medicine, University of Sassari, Via Vienna 2, 07100 Sassari, Italy; g.siddi1@phd.uniss.it (G.S.); mpmeloni@uniss.it (M.P.M.); mattiamigoni35@gmail.com (M.M.); d.cabras1991@gmail.com (D.C.); mrcuccu@uniss.it (M.C.); desantis@uniss.it (E.P.L.D.S.); scarano@uniss.it (C.S.); 2Department of Bacteria, Parasites & Fungi, Statens Serum Institut, Artillerivej 5, 2300 Copenhagen, Denmark; pgy@ssi.dk (P.G.); mtd@ssi.dk (M.T.); 3Department of Food Hygiene and Environmental Health, Faculty of Veterinary Medicine, University of Helsinki, FI-00014 Helsinki, Finland; maria.fredriksson-ahomaa@helsinki.fi

**Keywords:** zoonotic pathogens, whole genome sequencing, antibiotic resistance, wild game meat, carcass hygiene

## Abstract

The objective of this investigation was to evaluate *Salmonella* and *Yersinia enterocolitica* prevalence in wild boars hunted in Sardinia and further characterize the isolates and analyse antimicrobial resistance (AMR) patterns. In order to assess slaughtering hygiene, an evaluation of carcasses microbial contamination was also carried out. Between 2020 and 2022, samples were collected from 66 wild boars hunted during two hunting seasons from the area of two provinces in northern and central Sardinia (Italy). Samples collected included colon content samples, mesenteric lymph nodes samples and carcass surface samples. *Salmonella* and *Y. enterocolitica* detection was conducted on each sample; also, on carcass surface samples, total aerobic mesophilic count and *Enterobacteriaceae* count were evaluated. On *Salmonella* and *Y. enterocolitica* isolates, antimicrobial susceptibility was tested and whole genome sequencing was applied. *Salmonella* was identified in the colon content samples of 3/66 (4.5%) wild boars; isolates were *S. enterica* subs. *salamae*, *S.* ser. *elomrane* and *S. enterica* subs. *enterica*. *Y. enterocolitica* was detected from 20/66 (30.3%) wild boars: in 18/66 (27.3%) colon contents, in 3/66 (4.5%) mesenteric lymph nodes and in 3/49 (6.1%) carcass surface samples. In all, 24 *Y. enterocolitica* isolates were analysed and 20 different sequence types were detected, with the most common being ST860. Regarding AMR, no resistance was detected in *Salmonella* isolates, while expected resistance towards *β*-lactams (*blaA* gene) and streptogramin (*vatF* gene) was observed in *Y. enterocolitica* isolates (91.7% and 4.2%, respectively). The low presence of AMR is probably due to the low anthropic impact in the wild areas. Regarding the surface contamination of carcasses, values (mean ± standard deviation log_10_ CFU/cm^2^) were 2.46 ± 0.97 for ACC and 1.07 ± 1.18 for *Enterobacteriaceae*. The results of our study confirm that wild boars can serve as reservoirs and spreaders of *Salmonella* and *Y. enterocolitica*; the finding of *Y. enterocolitica* presence on carcass surface highlights how meat may become superficially contaminated, especially considering that contamination is linked to the conditions related to the hunting, handling and processing of game animals.

## 1. Introduction

The wild boar (*Sus scrofa*) is an ungulate belonging to the Suidae family and is one of the most common wildlife species; its diffusion is widespread due to the broad spectrum of habitat types that this species can inhabit, ranging from semi-arid environments to marshes, forests and grasslands [1]. The endemic species in Sardinia (Italy) is *Sus scrofa meridionalis*, classified as a separate subspecies [2]. The wild boar population has been steadily growing in Europe for several decades, with annual increases in population that can exceed 100 percent [3]. The warming temperatures, especially in winter; the high reproductive rate and adaptability of this species; the absence of natural predators; and the rural depopulation are among the reasons given for the constant increase in the wild boars’ population [4,5,6]. Due to the growing numbers, wild boars are considered an invasive species [7], and they directly and negatively affect the ecosystem in numerous ways, including preying on a wide range of vertebrate and invertebrate species, damaging nest sites and flora, outcompeting local wildlife and serving as disease vectors [8,9,10]. In particular relation to the latter, wild boars are an omnivorous species whose feeding behaviour determines the contact and consumption of different types of foods, of both vegetable and animal origin, including mammals and reptiles, and occasionally, they act as scavengers [11]. Moreover, it is increasingly common for wild boars to approach urban areas attracted to food subsidies in areas of high human activity and therefore come into contact with scraps and waste [12]; due to these reasons, wild boars can potentially come into contact with pathogenic microorganisms. In fact, numerous studies have underlined the importance of wild boars as reservoirs and spreaders of enteric and foodborne pathogens, such as *Salmonella* spp. and *Y. enterocolitica* [13,14,15,16]. 

The zoonotic risk derived from infected wildlife is associated with the spread of pathogens from indirect contamination (contact with farm animals and pastures) and the direct contamination of game meat [14]. In recent years, both the demand for wild boar meat and the consumption of game meat in general are rising; this trend is due to the characteristics of this type of meat, which complies with several consumer demands: it offers good sensory and nutritional profiles and has less of an environmental effect than farmed meat since it originates from animals that were born and grown in natural environments until the moment of harvest [17,18]. However, there are certain issues with game meat; in particular, the lack of microbiological standards and process hygiene criteria makes it challenging to organize supply chain controls. According to EC Regulation 853/2004 (Section 4, Chapter 1), it is sufficient for one person among a group of hunters to have knowledge of hygiene and proper handling techniques and laws concerning the conditions of meat and public health. Moreover, as muscle tissue is considered virtually sterile, many factors can affect the microbiological conditions of meat obtained from game animals, including the types of microorganisms carried by each species, the circumstances of harvest and the conditions under which the carcass is butchered, handled and stored [19,20]. In this framework, the microbiological quality and safety of game meat is highly dependent on the sanitary status of the hunted animals and on the slaughtering and meat processing environments and procedures applied [21,22]. These factors can result in the contamination of meat with consequent possible infection for consumers. 

Antimicrobial resistance (AMR) is a pressing worldwide issue: the loss of efficacy of antibiotics against common pathogens causes a significant clinical problem that has lately been compared to the challenges of climate change since it is a global-scale natural process that has been aggravated by human activity [23]. AMR has already been reported in bacteria of wildlife: although it is unlikely that wild animals come into contact with antibiotic substances, the overlap across habitats, in particular the interaction with humanized environments can cause the contact of wildlife with resistant bacteria from humans and other species [24]. In this regard, it has been reported that wildlife populations living in close proximity to humans exhibit higher levels of resistance, while populations living in natural and remote habitats show little to no resistance [25,26,27]. Wild boars, due to their ethological characteristics of being widespread and sharing habitat and distribution with other animals and humans have the potential to be reservoirs and spreaders of resistance genes, acting as a bridge between environments with a strong human influence and wild regions [28]. Numerous reviews have recently investigated the prevalence of AMR genes in wild boars in Europe, concluding that prevalence rates are highly variable and so are the antimicrobial susceptibility profiles and resistance genes detected [27,28,29,30,31].

To the best of our knowledge, very little is known regarding the microbiological quality and safety of meat and the AMR prevalence of pathogens from wild boars hunted in Sardinia; on this basis, the objective of this investigation was to evaluate the prevalence of *Salmonella* and *Y. enterocolitica* in wild boars hunted in Sardinia in order to characterize the isolates and evaluate the AMR pattern; moreover, in order to assess the slaughtering hygiene, an evaluation of the contamination of the carcasses at the end of the slaughter was carried out.

## 2. Materials and Methods

### 2.1. Sampling 

Between 2020 and 2022, samples were taken from 66 wild boars hunted from the area of two provinces, Sassari and Nuoro, in northern and central Sardinia (Italy). The animals were hunted during two consecutive hunting seasons. Wild boars were hunted by authorized hunting companies in the months defined by national and regional legislation (Region of Sardinia Decree number 7602/2011 of 24 August 2020 for season 2020–2021 and Decree number 846/13 of 23 August 2021 for season 2021–2022). A total of 21 wild boars were hunted in the 2020–2021 hunting season and 45 in the following hunting season. According to the hunting area, 53 animals were hunted in the Sassari province and 13 in the Nuoro province. 

Driven hunts were performed with hunting dogs during early mornings by parties of five to fifteen hunters; the hunters, armed with rifles, set up various positions, while beaters and hounds herded the boars in the direction of the guns. Carcasses were bled in the field, then the wild boars were collected in pick-up vehicles by the end of the hunting day. After harvesting, the carcasses were transported to dedicated facilities, “hunting houses”, where the slaughtering operations were carried out. Five hunting houses were utilized: three (A, B, C) were in the Sassari province and two (D, E) were in the Nuoro province. Figure 1 specifies the position of the hunting houses in Sardinia.

The following samples were taken from each wild boar immediately after evisceration:Mesenteric lymph nodes: at least 25 g of lymph nodes in the ileo-caecal regions were cut out with a sterile, disposable scalpel and collected in a sterile plastic bag (3M Health Care, Milan).Colon content: the colon was incised with a sterile, disposable scalpel and at least 25 g of its contents was collected in a sterile plastic bag.Carcass surface: samples were taken after evisceration by means of a non-destructive method with a sterile sponge pre-moistened with 10 mL of sterile buffered peptone water (BPW, 3M Health Care, Milan) at the following points: ham, loins, abdomen and throat; these points were selected as they are indicated by European legislation for pig carcasses at the slaughterhouse (Reg. CE No. 2073/2005; ISO 17604:2015 [32]). Sampling was carried out using the same sponge for the four points, with a sterile 10 × 10 cm^2^ delimiter (Copan, Brescia, Italy), proceeding from the least contaminated point (ham) to the most contaminated (throat). The sponges were handled with a sterile glove and placed inside sterile sponge bags. All the samples were transported to the laboratory at +4 °C and processed within 24 h after collection. A total of 181 samples were collected, divided into 66 colon content samples, 66 mesenteric lymph node samples and 49 carcass surface samples. Carcass surface samples were collected from wild boars slaughtered in hunting houses A, B and C; in the other hunting houses (D, E) the skinning was conducted the day after harvesting, and it was therefore not possible to evaluate the surface contamination of the carcass.

### 2.2. Microbiological Analysis

Medium for microbiology and reagents were purchased from Biolife (Italy). On each sample, *Salmonella* and *Y. enterocolitica* detection was conducted. For the detection of *Salmonella* spp., the ISO 6579:2020 [33] method was used. The species confirmation of the isolates was performed via the application of simplex PCR for the search of the *bcfC* gene according to the described protocol [34]. On *Salmonella* isolates phenotypic serotyping was conducted by slide agglutination and phase inversion (ISO 6579:2020) [33].

*Y. enterocolitica* detection was conducted according to ISO 10273:2017 [35], with modifications [36], as previously described [37]. At least five colonies with typical appearance were collected from each CIN agar plate and subjected to preliminary characterization tests (urea, sucrose, sorbitol) and biochemical confirmation in Klinger Iron Agar (KIA, Biolife). Species identification was performed via PCR through the amplification of a 330 bp fragment of the 16S rRNA gene [38]. The biotyping and serotyping of *Y. enterocolitica* strains was carried out at Statens Serum Institut: isolates were serotyped in the slide agglutination test with use of somatic antigens (SSI Diagnostica, Hillerød, Denmark). The strain was classified as nonidentified (NI) in the absence of agglutination with any of the sera. The biotype was determined based on indole production, and salicin, xylose and trehalose (SSI Diagnostica, Denmark) fermentation was carried out according to the ISO standard [35].

On carcass surface samples, decimal dilutions were prepared (ISO 6887:2017) [39] and total aerobic colony count (ACC) and *Enterobacteriaceae* count were conducted according to ISO 4833-1:2013 [40] and ISO 21528:2017 [41], respectively. As regards the level of surface contamination of the carcasses, the results were expressed as colony-forming units per square centimetre (CFU/cm^2^) on a logarithmic scale (log_10_) and compared with the process hygiene criteria thresholds established by the Regulation Commission (EC) No. 2073/2005 for pig carcasses, as the slaughtering procedure was similar to the one used for wild boars. The thresholds reported in the Regulation have been modified to adapt to the non-destructive method, as required by the Italian State-Regions agreement (41/2016): the m and M values indicated by the regulation, intended to distinguish between “satisfactory” (mean log CFU/cm^2^, m), “acceptable” (mean log CFU/cm^2^) and “unsatisfactory” (mean log CFU/cm^2^, M) results, were reduced by 20%.

### 2.3. Antimicrobial Susceptibility Testing

The antibiotic susceptibility of the isolates was tested using the Kirby–Bauer disc-diffusion method, according to the recommendations of the European Committee on antimicrobial susceptibility testing [42]. Mueller–Hinton agar (Microbiol, Italy) and commercial antimicrobial susceptibility discs (ThermoFisher Scientific, Waltham, MA, USA) were used. Plates were incubated at 35 ± 1 °C for 18–24 h. All isolates were tested for amikacin (Ak, 30 μg), ampicillin (Amp, 10 μg), amoxicillin/clavulanic acid (Aug, 20 μg and 10 μg, respectively), azithromycin (Azm, 15 μg), cephazolin (Kz, 30 μg), cefoxitin (Fox, 30 μg), ceftriaxone (Cro, 30 μg), cefotaxime (CTX, 30 μg), ceftazidime (Caz, 5 μg), ciprofloxacin (Cip, 5 μg), doxycycline (Do, 30 μg), imipenem (Ipm, 10 μg), kanamycin (K, 30 μg), levofloxacin (Lev, 5 μg), meropenem (Mem, 10 μg), nalidixic acid (Na, 30 μg), streptomycin (S10, 10 μg), sulphonamide (S, 300 μg), tetracycline (Te, 30 μg) and trimethoprim/sulfamethoxazole (Sxt, 1:19, 25 μg). According to the test results, isolates were categorized as susceptible or resistant according to the EUCAST recommendations, and intermediate isolates were considered susceptible. 

### 2.4. Whole Genome Sequencing 

Whole genome sequencing (WGS) was carried out at Statens Serum Institut on 3 *Salmonella* isolates and 25 *Y. enterocolitica* isolates. Sequencing data were uploaded to NCBI under BioProject PRJNA1043856. Bioproject accession number for all the isolates are listed in Appendix A. Genomic DNA was extracted from an enzymatic pre-lysis step prior to automated purification using the MagNA Pure 96 DNA and Viral NA Small Volume Kit and DNA Blood ds SV 2.0 protocol (Roche Diagnostics). Genomic libraries were constructed, and sequencing was performed using the Nextera XT Kit (Illumina) and 300-cycle kits on the NextSeq^®^ 550 (Illumina, San Diego, CA, USA) platform according to the manufacturer’s instructions. The quality control of the obtained sequencing data was conducted using Bifrost software (https://github.com/ssi-dk/bifrost, accessed on 31 July 2023) to ensure adequate sequencing depth, species verification and the identification of contamination issues. 

*Salmonella* serovars were detected from raw reads using SeqSero ver. 1.0 (for reference see http://denglab.info/SeqSero2, accessed on 31 July 2023) and subspecies were predicted using an in-house script based on the seven-gene MLST (multi-locus sequence type). MLST was determined using read mapping and named according to the Achtman seven-genes MLST scheme for *Salmonella* [43]. 

*Salmonella* and *Y. enterocolitica* genomes were uploaded to Enterobase [44,45]; on *Y. enterocolitica* genomes, seven-genes MLST and core-genome MLST (cgMLST) were applied [46]. The seven-gene MLST on *Yersinia* isolates was performed according to the McNally scheme [41]. The minimum spanning tree was generated with the minimal spanning tree algorithm MSTree V2 in EnteroBase. *Salmonella* and *Y. enterocolitica* genomes were subsequently compared to the publicly available genomes using the hierarchical clustering of cgMLST (HierCC clustering) at different levels of resolution [45].

On all samples, resistance, virulence and plasmid-associated genes were obtained from BioNumerics 8.1 plugin (Applied Maths, Sint Martems Latem, Belgium), AMRFinder [47], ResFinder [48], PlasmidFinder [49] and VirulenceFinder [50]. 

The results of the present study were compared with a similar previous investigation conducted on wild boars from the Asinara National Park [51].

### 2.5. Statistical Analysis 

Differences in the prevalence of *Salmonella* and *Y. enterocolitica* between samples (lymph nodes, colon content and carcass surface), hunting houses and hunting days were evaluated using one-way ANOVA with Tukey’s post hoc HSD. The significance level was defined as *p* < 0.05.

## 3. Results

### 3.1. Prevalence of Salmonella and Y. enterocolitica

*Salmonella* was identified in 3/66 wild boars, with a prevalence of 4.5%. The samples that tested positive were colon content samples only, with a prevalence of 4.5% (3/66) of the total colon content samples and a prevalence of 1.6% (3/181) of the total samples collected from wild boars (colon contents, lymph nodes, carcass surface). *Y. enterocolitica* had an overall prevalence of 30.3% (20/66) in wild boars. Among the *Y. enterocolitica*-positive animals, 2/20 (10%) were positive in colon content and carcass surface samples and 2 more animals tested positive in colon content and lymph nodes samples, while 1/20 (5%) were positive for *Y. enterocolitica* in carcass surface samples only. In particular, *Y. enterocolitica* was identified in 18/66 (27.3%) colon content samples, in 3/66 (4.5%) mesenteric lymph nodes samples and 3/49 (6.1%) carcass surface samples. Table 1 shows the results regarding the prevalence of *Salmonella* and *Y. enterocolitica* in wild boars and in the samples. 

Among the wild boars that tested positive for *Salmonella*, 2/3 were also positive for *Y. enterocolitica* in the intestine or lymph nodes. The prevalence of wild boars carrying *Salmonella* and *Y. enterocolitica* in relation to hunting houses is reported in Table 2; the highest number of wild boars positive for the two pathogens was detected in hunting house A, with a total of 12/28 positive individuals and a prevalence of 42.8%. However, no statistically significant difference (*p* > 0.05) was observed between the prevalence recorded in the hunting houses under investigation.

Overall, 3 *Salmonella* isolates and 24 *Y. enterocolitica* isolates were collected and submitted to further analysis.

### 3.2. Antimicrobial Susceptibility Testing

*Salmonella* isolates (3/3, 100%) were susceptible to all antimicrobials tested. Regarding *Y. enterocolitica* isolates, three different AMR profiles were identified: 10/24 (41.7%) showed resistance to amoxicillin-clavulanic acid, ampicillin and cefoxitin (AugAmpFox); 11/24 (48.8%) showed resistance to amoxicillin-clavulanic acid and ampicillin (AugAmp); 1/24 (4.2%) showed resistance to ampicillin (Amp) and 2/24 (8.3%) were sensible to all antimicrobials tested. Overall, 22/24 (91.7%) of *Y. enterocolitica* isolates showed phenotypic resistance to at least one beta-lactam compound. The AMR profile of *Salmonella* and *Y. enterocolitica* isolates is reported in Table 3. 

### 3.3. Salmonella Characterization

The three *Salmonella* isolates we found were classified as follows: (i) *S.* subsp. *salamae* (antigenic formula 48:h,z:1,5), (ii) *S.* ser. *elomrane* (9:z38:-) and (iii) a novel serotype of *S.* subs. *enterica* (28:e,h:z6). The third *Salmonella* strain identified is not currently referable to any known serotype and the strain has been sent to the European Union Reference Laboratory for *Salmonella* (EURL-*Salmonella*) for further study. Multilocus sequence types (ST) of the strains were, respectively, ST10546, ST7139 and ST10597, as reported in Table 3. The genetic characterization of genes in the virulence factors database showed that a total of 104 virulence genes were detected in the three *Salmonella* isolates. The *Salmonella* operons *bcfABCDEFG*, *csg(agf)ABCDEFG*, *fimCDFHI*, *invABEFGHJ*, and *sipABCD* were identified in all the isolates; moreover, *S. elomrane* also had the *lpfABCDE* operon. The following plasmids were detected: IncFIB(S) in ST 7138 strain, IncFII(S) in the novel serotypes. *Salmonella* isolates virulence genes detected are reported in Figure 2. 

Regarding AMR genes, no resistance genes were detected. 

### 3.4. Yersinia enterocolitica Characterization 

Among *Y. enterocolitica* isolates, 13/24 (54.2%) were biotype 1A and 11/24 (45.8%) were biotype 2. Biotype 2 strains were serotype O:3 (6/11, 54.5%) and O:5 (3/11, 27.3%); in 2/11 (18.2%), it was not possible to detect the serotype phenotypically. Among the isolates, 23 STs were detected. The single linkage tree in Figure 2 shows the genetic linkage between the isolates. The most common MLST sequence type was ST860, detected in five samples isolated from four wild boars hunted on two different days but in the same hunting house (A): the single linkage tree based on cgMLST (Figure 3) shows the genetic linkage between the isolates; regarding ST860 strains, isolates were genetically closely related with <7 SNP. 

Regarding virulence genes (Figure 4), all isolates had two or more virulence genes, namely *arsB* (24/24, 100%), *arsR* (23/24, 95.8%), *inv* (23/24, 95.8%), *myfA* (1/24, 4.2%), *yfeB* (24/24, 100%), *ymoA* (23/24, 95.8%) and *ystB* (1/24, 4.2%). Regarding AMR genes, two genes were detected, namely *blaA* and *vat(F)*. The *blaA* gene was detected in 100% (24/24) of the isolates. The *vat(F)* gene was found in the ST332 isolate (1/24, 4.2%). WGS also allowed the detection of five *Yersinia aleksiciae* isolates, which were detected in 5/66 (7.6%) wild boars. The isolates were found from colon content samples (3/66, 4.5%) and carcass surface samples (2/49, 4.1%) collected from hunting houses B and D. *Y. aleksiciae* isolates (5/5, 100%) had *arsB* and *ymoA* genes. No other typical *Y. enterocolitica* virulence or AMR genes were detected. 

### 3.5. Bacterial Contamination of Carcasses

As shown in Table 4, samples were taken over 12 hunting days and in three hunting houses (A, B and C). The ACC median value (log_10_ CFU/cm^2^) in hunting house A was 1.98, with a minimum of 1.98 and a maximum of 4.35; in hunting house B, the value was 2.47, with a minimum of 1.47 and a maximum of 3.91; and in hunting house C, the median was 3.37 (minimum of 2.93 and maximum of 4.00). Regarding *Enterobacteriaceae*, in hunting house A, the median value (log_10_ CFU/cm^2^) was 1.57, with the minimum value below the detection limit and the maximum value of 4.35; in hunting house B, the median value was 0.58 (minimum below the detection limit and maximum of 2.16); hunting house C had a median value equal to 2.48 (minimum of 1.60 and maximum of 3.48). In relation to the compliance with process hygiene criteria, the mean (log_10_ CFU/cm^2^ ± standard deviation) values for ACC and *Enterobacteriaceae* were 2.46 ± 0.97 and 1.07 ± 1.18, respectively. In particular, 37/49 (75.5%) of the samples taken from the surface of the carcasses showed CCA values <3.2 log_10_ CFU/cm^2^ and were considered satisfactory, 8/49 (16.3%) samples had values between 3.2 and 4 log_10_ CFU/cm^2^ were considered acceptable and 4/49 (8.2%) samples had values >4 log_10_ CFU/cm^2^ and were considered unsatisfactory. For the *Enterobacteriaceae* count, 30/49 (61.2%) samples showed values <1.6 log_10_ CFU/cm^2^ and were considered satisfactory, 9/49 (18.4%) samples had values between 1.6 and 2.4 log_10_ CFU/cm^2^ were considered acceptable and 10/49 (2%) samples had values >2.4 log_10_ CFU/cm^2^ and were considered unsatisfactory. 

## 4. Discussion

The prevalence of carrier wild boars (positive in the colon content and/or mesenteric lymph nodes) was higher for *Y. enterocolitica* (30.3%) than for *Salmonella* (4.5%). Two cases of co-infection were recorded: *Y. enterocolitica* isolates were also found in 2/3 of wild boars testing positive for *Salmonella*. Overall, 33.4% (22/66) of the wild boars carried at least one of the considered pathogens. 

In this investigation, *Salmonella* was isolated in 4.5% (3/66) of wild boars sampled; a similar prevalence has been identified in surveys conducted in central and northern Italy: 4.2% in Tuscany and 6% in Liguria [52,53]. Other authors in various Italian regions have identified a higher prevalence, with a range from 10.8% to 17% [30,54,55,56]. Similar surveys in European countries have shown prevalence ranging from 7.7% to 22% [13,57,58,59,60]. However, prevalence studies conducted on hunted wildlife have limitations because the animals sampled may not be representative of the entire population. This is due to hunting strategies, which tend to select older and heavier animals that are less likely to be *Salmonella* carriers and shedders than younger ones. In fact, significant differences regarding *Salmonella* prevalence were observed in pigs of different ages, with pathogen elimination rates decreasing with age [61,62,63]; this may depend on a greater susceptibility to infection in young pigs [64], associated with a greater ability of the pathogen to establish infection in the early stages of life [63]. In this framework, the prevalence of the pathogen in wild populations can be underestimated [56].

In a previous investigation [51] conducted on wild boars coming from the Asinara Island National Park (northwestern end of Sardinia), the prevalence of *Salmonella* was 46.7%, notably much higher than the findings of the present study. A possible explanation for this difference may depend on the fact that the wild boars were periodically captured within the Asinara Park, as part of the specific plan for the numerical control of the wild boar population on the island. The animals were transported to a slaughterhouse in Sardinia and were therefore possibly affected by the same stressors (capture, handling and transport), which are known to increase the susceptibility and probability of spreading *Salmonella* in pigs subjected to slaughter [65]. On the other hand, the wild boars in the present study did not undergo the cited stressors before the moment of harvest. 

Two *Salmonella* species and three different serotypes were identified: (i) *S.* subsp. *salamae*, (ii) *S. elomrane* and (iii) a novel serotype of *S.* subs. *enterica* (28:e,h:z6). *S. salamae* and *S. elomrane* strains are not common in human outbreaks; comparisons with public databases (Enterobase) give few results and also highlights the limited diffusion of these serotypes. The novel serotype did not yield any results regarding publicly available similar isolates. The genome of *S. salamae* ST10546 was compared to other *Salmonella* genomes using the hierarchical clustering of cgMLST (HierCC clustering) on Enterobase; the isolate belonged to HC400_202057, which also included six genomes from strains isolated from wild boars in Sardinia in 2016 and 2019 [51]. No other genomes were assigned to cluster HC200_359455, which means that no genome in EnteroBase has less than 200 allele differences, indicating that the *S. salamae* ST10546 is a rather rare type and, to the best of our knowledge, currently identified only in wild boars in Sardinia. Interestingly, also *S. elomrane* isolates have been previously isolated in Sardinia from wild boars of the Asinara National Park and in both investigations possessed the same ST 7139. This finding is an indication of the circulation of these strains in wild environments. *S. salamae* and *S. elomrane* have also been linked to wild boars in other parts of Italy and Spain [30,51,60]. *S. enterica* and “non-*enterica*” subspecies are typically found in the environment and cold-blooded animals: subs. *salamae* strains have been detected in reptiles [64,66,67,68]; *elomrane* is a rare serotype and spread by reptiles or migratory birds is also suggested [69,70], although little information regarding this serotype is available. In the wild, the ingestion of contaminated food or water is considered the most typical transmission path of *Salmonella* [66]; therefore, the observation of *Salmonella* at colon content level and not in lymph node samples suggests that the infection of wild boars with *Salmonella* is linked to the diet. The main route of infection for humans is the consumption of the meat of infected animals or the contact (direct or indirect) with reptiles, particularly when they are kept as pets [71,72,73].

The genetic characterization of genes showed that the *Salmonella* operons *bcfABCDEFG*, *csg(agf)ABCDEFG*, *fimCDFHI*, *invABEFGHJ*, and *sipABCD* were identified in all the isolates; moreover *S. elomrane* also had the *lpfABCDE* operon. These cited operons are invasion-related genes, widespread in *Salmonella* isolates and typical of *S.* Typhimurium and are implicated in the colonization of intestinal tissues [74,75,76,77]. Also, the mig-14 and *mgtC* genes, related to *Salmonella* survival and proliferation in macrophages and host [78], were detected in *S. elomrane* and in the novel *Salmonella* strain. On the other hand, the *spv* operon, involved in the modulation of the host immune response to infection [74], was not detected in any strain. The novel *Salmonella* strains also showed the ratB gene, which is involved in long-term intestinal persistence encoding a Peyer’s patch and cecum colonization factor, and the *orgA* gene, which is involved in promoting cellular invasion of the pathogen [79,80,81]. *S. elomrane* and *S.* subsp. *salamae* had the *cdtB* gene, which encodes a variant of the cytolethal distending toxin (CDT), an important virulence factor for *S.* Typhi but is commonly found also in non-typhoidal serovars [82]. 

As regards *Y. enterocolitica*, it was isolated from 30.3% (20/66) of the wild boars sampled in our investigation; this prevalence is higher than observed in similar surveys in Italy that reported values between 2.9% and 17.8% [31,52,83]. Other authors have also reported a highly variable prevalence in wild boars in Europe, ranging from 1.3% to 33.3% [58,84,85,86]. The high prevalence of this microorganism in wild boars probably depends on the contact with other infected wild species and/or livestock reared in extensive grazing systems [9]; in particular, sheep have been described as a reservoir of *Y. enterocolitica* strains [87,88]. However, there is little information regarding *Yersinia* infections in sheep in Sardinia or regarding their relationship with species identified in wild boars, although this zoonotic pathogen has been already identified in raw sheep milk and cheese-making plants in Sardinia [37,89]. It is also noteworthy that the wild boar sampling took place over two hunting seasons conducted from November to January over the course of two consecutive seasons (2020–2021 and 2021–2022). In this regard, some authors have observed a seasonality in the prevalence of pathogenic microorganisms, including *Yersinia*, in wild animals, with an increase in cases concentrated in the winter months [87,90]. Latent infection can, in fact, manifest itself in stressful conditions, such as those observed in the cold season in which animals, especially wild species, are exposed to low temperatures and food shortage [91]. The sequence types identified in *Y. enterocolitica* strains were very diverse. The most common sequence type was ST860, detected in five genetically closely related isolates found in four wild boars, processed in the same hunting house (A) and less than one month apart. As shown in Figure 2, overall, the ST860 strains had less than 10 SNP between each other. Although these data cannot distinguish between contaminations linked to animal-to-animal transmission or common environmental sources, genetic relatedness points to epidemiological connections among the strains. 

Regarding the virulence genes detected in the isolates, *inv*, *myfA* and *ystB* are chromosomal virulence genes that encode for the internalization factor invasin *invA* and the mucoid *Yersiniae* factor *myfA*, respectively [87]. The heat-stable enterotoxin gene *ystB* was detected in 1/24 (4.2%) of the strains, while the heat-stable enterotoxin gene *ystA* was never detected. Moreover, the *ymoA* gene was also detected in 23/24 (95.8%) of the strains, which negatively modulate the expression and transcription of virulence factors, *inv* and *yst* genes in particular [92,93]. The *ars* operon confers virulence and resistance to arsenite, arsenate and antimonite in *Yersinia* species; *yfe* is a transport system that accumulates both iron and manganese [94]. *YstB* gene is usually carried by 1A biotype strains, while *ystA* is detected in pathogenic 1B strains [43,87,90]; therefore, our findings suggest that the *Y. enterocolitica* strains detected in the present investigation are not particularly pathogenic.

Five *Y. aleksiciae* strains were isolated among wild boars. *Y. aleksiciae* is categorized among the *Y. enterocolitica*-like species [95,96]. Strains appear to have scarce pathogenicity, due to the lack of typical *Y. enterocolitica* virulence genes. The species seem to be well adapted to warm-blooded animals, and it has been isolated from the faeces of humans, rats, reindeer and pigs, as well as from dairy products [97]. However, the importance of this species must be studied further. 

As reported in Table 3, in our study, no AMR profile was detected in *Salmonella* isolates. *Y. enterocolitica* strains (24/24, 100%) possessed the *blaA* gene, which encodes for the production the *β*-lactamase BlaA (a constitutive class A enzyme) [98]. The presence of the gene was reflected in phenotypical resistance to at least one *β*-lactam compound in 91.7% (22/24) of the strains. This result was expected as the *blaA* gene has been reported widely in *Y. enterocolitica* isolates, regardless of biovars or the geographical origin of the strains, and intrinsic resistance to *β*-lactam compounds has been suggested by EUCAST [42]. In 1/24 (4.7%) *Y. enterocolitica* isolate (ST332), the *vat(F)* gene was detected; it is a chromosomal gene that encodes resistance towards streptogramin and is also widespread in *Y. enterocolitica* strains [99,100]. The detection of resistance patterns and genes are typical of the species and are widespread in the isolates, suggesting that these are inherent resistances, and most likely not acquired from the environment or from contact with other resistant strains. The low prevalence of AMR in both *Salmonella* and *Y. enterocolitica* isolates is probably due to the low selective pressure given by the low level of exposure to antimicrobial substances of resistant microbial populations. This positive result could also indicate the scarce anthropic impact in the areas where wild boars live [27]. 

Regarding surface contamination of carcasses, the mean values (log_10_ CFU/cm^2^ ± standard deviation) of the ACC were 2.46 ± 0.97, while the mean values of *Enterobacteriaceae* were 1.07 ± 1.18. In particular, based on the limits established for surface samples of pig carcasses by EC Regulation no. 2073/2005 and the Italian State-Regions Agreement 41/2016, 8.2% of the samples (4/49) and 2% (10/49) showed values higher than the maximum limit established for CCA and *Enterobacteriaceae*, respectively. The average values observed in our investigation are, however, lower than those reported in similar studies [22,55,101,102]. The average values identified in the present study indicate an overall correct application of hygiene practices (GHP) and slaughtering practices (GMP) during game meat handling. However, the values differed in the sampling days, particularly for the levels of *Enterobacteriaceae* (*p* < 0.05), where the range was between a minimum mean value of 0.10 ± 0.22 and a maximum mean value of 3.68 ± 0.52. The production of meat from hunted wildlife might occasionally reveal some deficiencies in the GHP and GMP application process. Particularly, the harvesting and processing of meat frequently take place in conditions that are unsuitable for meat production [22]. In this regard, in our investigation, after killing, the wild boars were collected on pick-up vehicles, and therefore exposed to environmental conditions, until the end of the hunting day; subsequently, the carcasses were transported to dedicated facilities where the slaughter operations were carried out. At these stages, the time and conditions between the moment of death and processing were highly variable among animals [20]. Some authors have observed the higher contamination of carcasses on hunting days characterized by adverse weather conditions, which provide settings for faecal and environmental contamination or the spread of pre-existing contamination [22]. In the present investigation, 6.1% (3/49) of the carcass surface samples tested positive for *Y. enterocolitica* and one wild boar was positive only in carcass surface samples. This is probably due to cross-contamination and poor hygiene in the processing phases or incorrect evisceration practices. The presence of wild boars excreting *Salmonella* and *Y. enterocolitica* strains poses a concern to consumers since it is feasible that meat and carcasses become superficially contaminated, particularly since the conditions during the hunting, handling and processing of game carcasses are crucial to microbiological contamination. Even though game meat is usually consumed cooked, wild boar meat products are frequently only dry-cured (such as traditional dry, fermented sausages), and it is therefore possible for pathogenic bacteria to cause foodborne infections [101]. The education of hunters and the use of appropriate hygienic procedures are essential in this circumstance.

## 5. Conclusions

Monitoring enteric pathogens in wildlife is crucial to trace the evolution and factors contributing to selection and spread. The results of our investigation confirm that wild boars in Sardinia can act as reservoirs and spreaders of both *Salmonella* and *Y. enterocolitica* strains. The wild boars in this study hosted uncommon strains, due to the wild environment. The data show overall good carcass hygiene status, with generally acceptable contamination levels. However, the presence of wild boars carrying enteric pathogens represents a risk for the consumer as it is possible that the superficial contamination of carcasses and meat may occur, especially throughout the processing of game meat. In this context, in particular, the training of hunters and the application of good hygiene practices is fundamental. Regarding antimicrobial resistance, an overall low prevalence of resistant strains in the isolates identified from wild boars hunted in Sardinia, with the detection of only intrinsic and expected resistance profiles. This positive result could indicate the low level of exposure to antimicrobial compounds and scarce contact with resistant strains in wild areas. However, the constant numerical increase of the wild boar population and the possible contacts between humans and farmed and wild animals due to the expansion of urban areas make it necessary to constantly monitor the spread of antimicrobial resistance in wild species.

## Figures and Tables

**Figure 1 foods-13-00065-f001:**
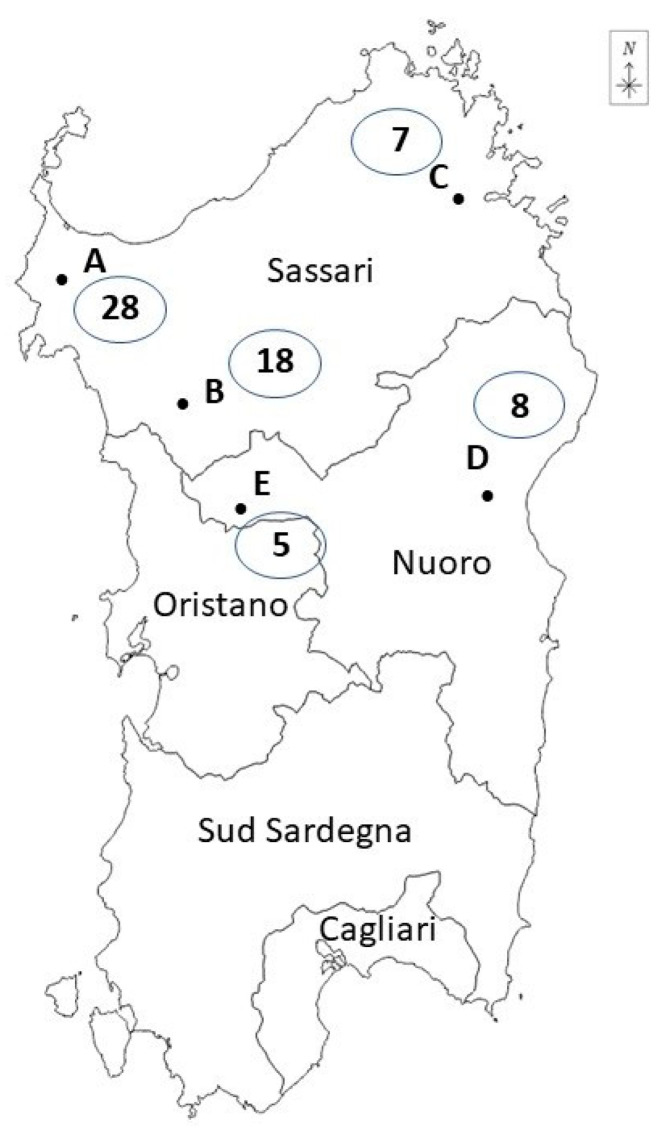
The geographical distribution of the hunting houses included in the study (Sardinia, Italy) and the number of wild boars sampled.

**Figure 2 foods-13-00065-f002:**
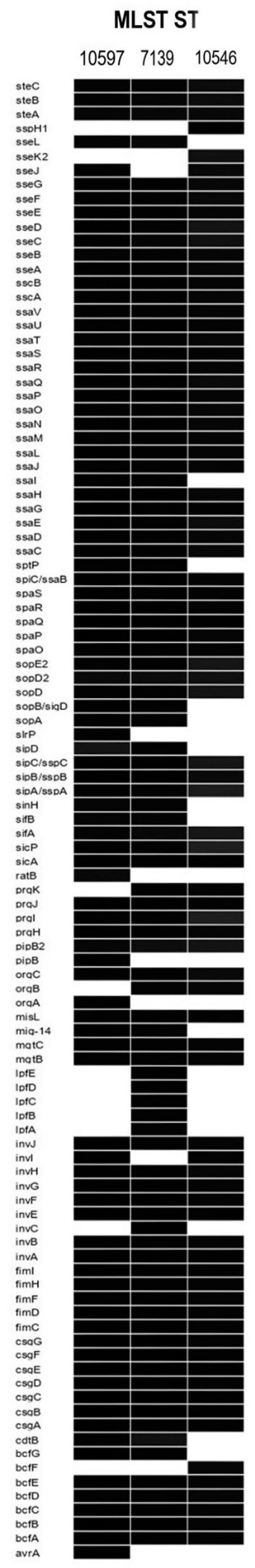
Virulence genes of *Salmonella* strains.

**Figure 3 foods-13-00065-f003:**
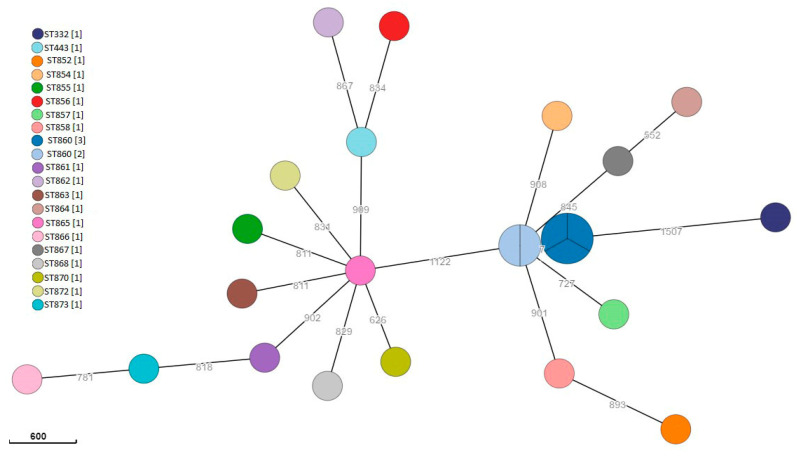
Minimum spanning tree of *Y. enterocolitica* isolates based on cgMLST. Branch labels indicate allelic distances. Numbers in square brackets indicate the number of isolates for each ST.

**Figure 4 foods-13-00065-f004:**
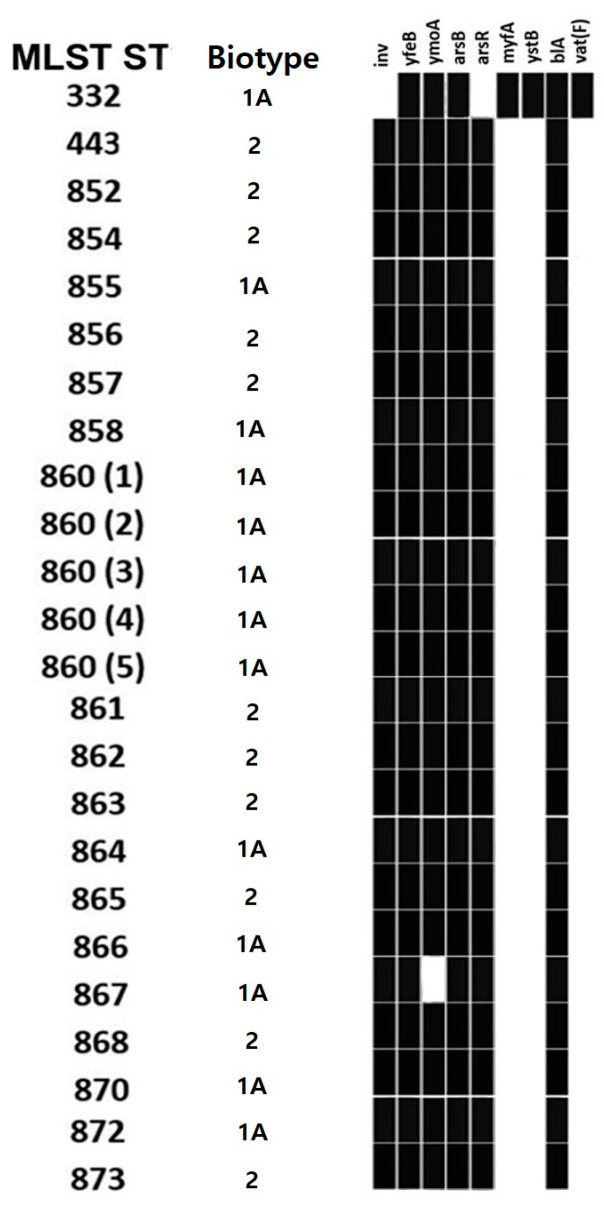
Virulence genes of *Y. enterocolitica* isolates based on MLST sequence type and biotype.

**Table 1 foods-13-00065-t001:** Prevalence of *Salmonella* and *Y. enterocolitica* in wild boars and samples of colon content, mesenteric lymph nodes and carcass surface (positive/total, prevalence %).

Pathogen	Tested Animals	Positive Animals	Positive Samples			
			Colon Content	Mesenteric Lymph Nodes	Carcass Surface	Total
*Salmonella*	66	3	3/66 (4.5)	0/66	0/49	3/181 (1.6)
*Y. enterocolitica*	66	20	18/66 (27.3)	3/66 (4.5)	3/49 (6.1)	24/181 (13.2)

**Table 2 foods-13-00065-t002:** Prevalence of wild boar carriers of *Salmonella* and *Y. enterocolitica* in terms of the hunting houses.

Hunting Houses	Sampled Wild Boars	Total Positive Wild Boars	Pathogen	
			*Salmonella*	*Y. enterocolitica*
A	28	12	2	10
B	18	7	1	6
C	7	1	0	1
D	8	2	0	2
E	5	1	0	1

**Table 3 foods-13-00065-t003:** AMR profile of *Salmonella* and *Y. enterocolitica* isolates.

Microorganism	MLST ST	Phenotypic Profile	Genotypic Profile
*Salmonella*	10546	ND	-
	7139	ND	-
	10597	ND	-
*Y. enterocolitica*	332	Amp Aug	*blaA*, *vat(F)*
	443	Amp Aug Fox	*blaA*
	852	Amp Aug	*blaA*
	854	Amp Aug	*blaA*
	855	Amp Aug	*blaA*
	856	Amp	*blaA*
	857	ND	*blaA*
	858	Amp Aug	*blaA*
	860 (1)	Amp Aug Fox	*blaA*
	860 (2)	Amp Aug	*blaA*
	860 (3)	Amp Aug	*blaA*
	860 (4)	Amp Aug	*blaA*
	860 (5)	Amp Aug Fox	*blaA*
	861	Amp Aug Fox	*blaA*
	862	Amp Aug Fox	*blaA*
	863	Amp Aug	*blaA*
	864	Amp Aug Fox	*blaA*
	865	ND	*blaA*
	866	Amp Aug Fox	*blaA*
	867	Amp Aug	*blaA*
	868	Amp Aug	*blaA*
	870	Amp Aug Fox	*blaA*
	872	Amp Aug Fox	*blaA*
	873	Amp Aug Fox	*blaA*

ND: no phenotypic resistance detected.

**Table 4 foods-13-00065-t004:** Bacterial contamination of carcasses during the sampling days (mean log_10_ CFU/cm^2^) in hunting houses A–C.

	Sampling Days	1	2	3	4	5	6	7	8	9	10	11	12
	Hunting Houses	A	A	B	A	C	A	B	A	B	A	B	B
	Number of Samples	7	1	8	8	3	5	2	6	4	1	2	2
ACC	Mean	1.61	1.33	2.68	2.85	3.43	1.69	2.83	2.48	2.70	2.87	3.12	1.85
	Median	1.50	1.33	2.50	3.02	3.37	1.32	2.83	1.96	2.61	2.87	3.21	1.85
	Minimum value	1.19	-	1.78	1.33	2.93	1.07	1.97	1.11	1.81	-	2.51	1.47
	Maximum value	2.14	-	3.83	4.15	4.00	2.89	3.68	4.35	3.78	-	3.91	2.23
*Ent*	Mean	0.58	0.15	0.89	3.68	2.52	0.10	0.84	1.52	0.57	0.30	0.77	0.00
	Median	0.33	0.15	0.61	4.00	2.48	0.00	0.84	1.53	0.29	0.30	0.77	0.00
	Minimum value	0.00	-	0.00	2.67	1.60	0.00	0.15	0.00	0.00	-	0.67	-
	Maximum value	2.01	-	2.16	4.00	3.48	0.51	1.53	2.87	1.70	-	0.87	-

ACC: aerobic colony count, *Ent*: *Enterobacteriaceae.*

## Data Availability

Sequencing data were uploaded to NCBI under BioProject PRJNA1043856. Bioproject accession number for all the isolates are listed in Appendix A.

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
