# Peer review of "Hunted Wild Boars in Sardinia: Prevalence, Antimicrobial Resistance and Genomic Analysis of Salmonella and Yersinia enterocolitica"

_foods, 2023, doi:10.3390/foods13010065_

Round 1

Reviewer 1 Report

Comments and Suggestions for Authors

Review of the manuscript no. 2761112 for Foods. Title “Hunted wild boars in Sardinia: prevalence, antimicrobial resistance and genomic analysis of Salmonella and Yersinia enterocolitica:

This is a well written manuscript that can be reconsidered when the authors have considered and corrected it according to the following minor comments/questions:

1.      Abstract lines 34-35 (and the same comment for Conclusions line 533 and for Discussion lines 518-519): Could the authors please re-phrase the sentence about “peculiar circumstances”, it sounds strange when you read it. A suggestion of new wording is “particularly since the conditions during hunting, handling and processing of game carcasses have been identified as crucial to microbiological contamination”.

2.      Introduction line 55: Please change the word “carrions”, it is very unusual. I suppose the authors mean that the wild boars eat everything including dead animals, i.e. thay are scavengers?

3.      Material and methods line 113: It would be interesting to know how the animals were killed (rifle?), and also what hunting method that was used (were dogs involved? , hunting during the day or night? et.c) and finally were the carcasses bled in the field? The authors are recommended to include some information about this in the paragraph 2.1 Sampling.

Comments on the Quality of English Language

Please see comments above.

Author Response

Revisions to manuscript “foods-2761112”:  Hunted wild boars in Sardinia: prevalence, antimicrobial resistance and genomic analysis of Salmonella and Yersinia enterocolitica

The modifications and improvements suggested by the reviewers have been implemented and included in the text. The table below displays the precise modifications based on the individual comments. Revisions in the text are highlighted with a different color (green for Reviewer 1 and blue for Reviewer 2).

Reviewer 1 suggestions

Modifications in the text (green)

Abstract lines 34-35 (and the same comment for Conclusions line 533 and for Discussion lines 518-519): Could the authors please re-phrase the sentence about “peculiar circumstances”, it sounds strange when you read it. A suggestion of new wording is “particularly since the conditions during hunting, handling and processing of game carcasses have been identified as crucial to microbiological contamination”.

Phrases have been modified accordingly in lines 34-35 (abstract), 520-522 (discussion) and 535 (conclusions).

Introduction line 55: Please change the word “carrions”, it is very unusual. I suppose the authors mean that the wild boars eat everything including dead animals, i.e. they are scavengers?

Yes, this is the intended meaning. “Carrions” has been deleted and the paragraph has been modified accordingly in lines 53-56.

Material and methods line 113: It would be interesting to know how the animals were killed (rifle?), and also what hunting method that was used (were dogs involved? , hunting during the day or night? et.c) and finally were the carcasses bled in the field? The authors are recommended to include some information about this in the paragraph 2.1 Sampling.

The wild boar hunts were driven hunts performed during early mornings by groups of 5-15 hunters where armed hunters (rifles) are placed on different stands and the wild boars are driven towards the guns by beaters and dogs. Information regarding the hunting technique has been added to the text in lines 113-116.

Reviewer 2 Report

Comments and Suggestions for Authors

I thoroughly enjoyed reading the manuscript entitled “Hunted wild boars in Sardinia: prevalence, antimicrobial resistance and genomic analysis of Salmonella and Yersinia enterocolitica”.

Here are my minor comments:

1. Wild boars carries several pathogenic bacteria however, 3 most important zoonoses are salmonellosis, yersiniosis and listeriosis, why did the authors leave Listeria isolates in this study?

2. It would be nice if the authors include a Figure showing the graphical distribution of the sampling area.

3. Can the authors specify what is N. or n. in lines 110, 187 and so on? Does it mean Nuoro or number? Why is it not specified in the supplementary table?

4. Please release the genome sequence of the isolates before the article is published.

5. Did the authors find co-infection with any other microbial pathogen other than Salmonella and Yersinia?

Author Response

Revisions to manuscript “foods-2761112”:  Hunted wild boars in Sardinia: prevalence, antimicrobial resistance and genomic analysis of Salmonella and Yersinia enterocolitica

The modifications and improvements suggested by the reviewers have been implemented and included in the text. The table below displays the precise modifications based on the individual comments. Revisions in the text are highlighted with a different color (green for Reviewer 1 and blue for Reviewer 2).

Reviewer 2 suggestions

Modifications in the text (blue)

Wild boars carries several pathogenic bacteria however, 3 most important zoonoses are salmonellosis, yersiniosis and listeriosis, why did the authors leave Listeria isolates in this study?

The observation is very accurate and appreciated. We did not search for Listeria in this study, however we will make sure to look for this pathogen in future studies on hunted wild boars. Nevertheless, we simultaneously investigated pathogenic E. coli in wild boar samples, the findings of which will be published in a future publication.

It would be nice if the authors include a Figure showing the graphical distribution of the sampling area.

Figure 1 indicating the geographical position of the hunting houses has been added to the text (lines 120-126).

Can the authors specify what is N. or n. in lines 110, 187 and so on? Does it mean Nuoro or number? Why is it not specified in the supplementary table?

N. was an abbreviation of “number”, however, it has been removed from the text since it may cause misunderstanding.

Please release the genome sequence of the isolates before the article is published.

The date of release of the genome sequence will be modified and brought forward.

Did the authors find co-infection with any other microbial pathogen other than Salmonella and Yersinia?

Yes, co-infection with Yersinia and pathogenic E. coli (UPEC and ExPEC) has been observed in two wild boars. Results of pathogenic E. coli investigation will be published soon.

Reviewer 3 Report

Comments and Suggestions for Authors

Line 23: did you mean S. elomrane ?

Line 28: β-lactams (use italic text format for β)

Line 36: use a semicolon to separate keywords

Line 49: use [4–6] instead of [4, 5, 6].

Line 52: use [8–10] instead of [8, 9, 10].

Line 61: use [13–16] instead of [13, 14, 15, 16].

Line 68: use [17,18] instead of [17, 18].

Line 77: remove space [19,20].

Line 80: remove space [21,22].

Line 40-81: In common scientific articles, maximum paragraphs between 15-22 lines are recommended, it is necessary to follow the recommendation

Line 91: use [25–27] instead of [25, 26, 27].

Line 97: use [27–31] instead of [29, 30, 27, 31, 28].

Line 107: rewrite… 2.1. Sampling

Line 108-112: Do you have any number of permits or authorizations to carry out the hunting of these animals? Could you include them?

Line 109: Mesenteric

Line 121: container.

Line 122: Colon

Line 123: bag.

Line 123: include the brand of the sterile bags used

Line 124: Carcass

Line 125: use mL instead of ml

Line 130: cm2

Line 141: rewrite… 2.2. Microbiological Analysis

Line 144: rewrite… to the described protocol [32]. On…

Line 148: Avoid using the authors' last names. According to the author guide, it is only necessary to use the corresponding reference number in square brackets.

Line 158-159: include the reference number used in brackets

Line 171: rewrite… 2.3. Antimicrobial Susceptibility Testing

Line 175: At the beginning of the materials and methods section, include a section on reagents where each of the reagents used in this research was acquired from.

Line 176: 18–24 h.

Line 176: 35 ± 1 °C

Line 186: rewrite… 2.4. Whole Genome Sequencing

Line 216: rewrite… 2.5. Statistical Analysis

Line 223: rewrite… 3.1. Prevalence of Salmonella and Y. enterocolitica

Line 246: use Y. enterocolitica

Line 247: insert a dot… %).

Line 256: (P>0.05)

Line 266: rewrite… 3.2. Antimicrobial Susceptibility Testing

Line 275: Table 3.

Line 275: insert a dot… isolates.

Line 279: rewrite… 3.3. Salmonella Characterization

Line 281: S. elomrane

Line 294: Figure 1.

Line 294: insert a dot… strains.

Line 297: rewrite… 3.4. Yersinia enterocolitica Characterization

Line 307: Figure 2.

Line 335: Figure 3.

Line 336: insert a dot… biotype.

Line 338: rewrite… 3.5. Bacterial Contamination of Carcasses

Line 340: remove space through the manuscript for CFU/cm2

Line 354: remove space… <1.6

Line 356: remove space… >2.4

Line 359: Table 5.

Line 360: insert a dot… A-C.

Line 370: remove space [46,47];

Line 371: use [30, 48–50] instead of [30, 48, 49, 50].

Line 372: use [13, 51–54]

Line 378: use [55–57]

Line 394: S. elomrane

Line 407: remove space… [30,45,54].

Line 409: use [60–62]

Line 410: remove space… [63,64].

Line 416: use [66–67]

Line 420: S. elomrane (correct through the manuscript if is correct)

Line 422: use [68–71]

Line 429: [74,75].

Line 434: avoid authors' last names

Line 436: use [52,78–80]

Line 439: [81,82].

Line 442: [34,83].

Line 447: [80,84].

Line 463: [86,87].

Line 466: [37,81,84].

Line 477: β-lactamase

Line 478,481: β-lactam

Line 484: [93,94].

Line 498: [22,49,95,96].

Line 501: did you mean (P<0.05), ?

Line 558: remove space… D.M.;

Line 574: S.M.;

Line 577: M.N.;

Line 577: S.A.

Line 589: F.J.M.;

Line 591: V.R.;

Line 591: R.A.;

Line 591: A.M.

Note: remove spaces between abbreviations of author names in references where required

Line 601: hygiene challenge?

Line 603: game meats.

Line 606: Hunted wild boar carcass hygiene: Roles of different factors involved in the harvest phase

Line 623: use italic text format for scientific names

Line 631: Yersinia enterocolitica

Line 636: Int. J.

Line 637: 32–35 ?

Line 639: Int. Dairy J.

Line 652: 138–152.

Line 650: Is the text format of the title correct?

Note: carefully review the format of each of the bows according to the authors' guide
